# The last common ancestor of animals lacked the HIF pathway and respired in low-oxygen environments

Daniel B Mills[1†], Warren R Francis[2†], Sergio Vargas[2], Morten Larsen[1], Coen PH Elemans[1], Donald E Canfield[1], Gert Wörheide[2,3,4]*

[1]Department of Biology, University of Southern Denmark, Odense, Denmark; [2]Paleontology & Geobiology, Department of Earth and Environmental Sciences, Ludwig-Maximilians-Universität München, Munich, Germany; [3]GeoBio-Center, Ludwig-Maximilians-Universität München, Munich, Germany; [4]SNSB - Bayerische Staatssammlung für Paläontologie und Geologie, Munich, Germany

**Abstract** Animals have a carefully orchestrated relationship with oxygen. When exposed to low environmental oxygen concentrations, and during periods of increased energy expenditure, animals maintain cellular oxygen homeostasis by enhancing internal oxygen delivery, and by enabling the anaerobic production of ATP. These low-oxygen responses are thought to be controlled universally across animals by the hypoxia-inducible factor (HIF). We find, however, that sponge and ctenophore genomes lack key components of the HIF pathway. Since sponges and ctenophores are likely sister to all remaining animal phyla, the last common ancestor of extant animals likely lacked the HIF pathway as well. Laboratory experiments show that the marine sponge *Tethya wilhelma* maintains normal transcription under oxygen levels down to 0.25% of modern atmospheric saturation, the lowest levels we investigated, consistent with the predicted absence of HIF or any other HIF-like pathway. Thus, the last common ancestor of all living animals could have metabolized aerobically under very low environmental oxygen concentrations.
DOI: https://doi.org/10.7554/eLife.31176.001

*For correspondence: woerheide@lmu.de

[†]These authors contributed equally to this work

Competing interests: The authors declare that no competing interests exist.

## Introduction

Crown-group animals (metazoans) originated around 850 million years ago (Ma) (*Dohrmann and Wörheide, 2017*), when atmospheric oxygen ($O_2$) was likely <10% of its present level (*Lenton and Daines, 2017*). Less atmospheric oxygen would had led to extensive deep-ocean anoxia and high redox variability in marine surface waters (*Reinhard et al., 2016*), thereby placing energetic constraints on the diversity, abundance, and physiology of heterotrophic eukaryotes, particularly animals (*Fenchel and Finlay, 1995*). In all previously examined animal lineages, decreasing cellular oxygen concentrations, whether environmentally or metabolically driven, alter gene expression via the hypoxia-inducible factor (HIF), a key pathway for maintaining physiological oxygen homeostasis (*Loenarz et al., 2011*).

The HIF pathway consists of two bHLH-PAS-domain transcription factors (HIFa and HIFb/ARNT) that form a heterodimer to initiate transcription when oxygen concentrations at the cellular level are low (*Semenza, 2007*). At higher oxygen levels, HIFa is hydroxylated by a member of the 2-oxoglutarate-dependent oxygenase family, HPH (*Bruick and McKnight, 2001*), also called PHD or EGL9, and is then signaled for destruction by the von Hippel-Lindau (VHL) protein, thereby avoiding its dimerization with HIFb and subsequent transcriptional activation (*Min et al., 2002*). Proteins in the HIF pathway have been characterized in many metazoan groups (*Kaelin and Ratcliffe, 2008*), including placozoans (*Loenarz et al., 2011*) and cnidarians (*Wang et al., 2014*), but have not been evaluated

**eLife digest** Almost all animals need oxygen to live. This is because they use oxygen to release much of the energy locked up in their diets. Oxygen may have also played a crucial role in the early evolution of animal life. Animals evolved from single-celled ancestors in the ocean over 800 million years ago. Before then, it is debated whether the atmosphere and ocean had enough oxygen to permit animals to evolve.

Oxygen levels are much higher now, but oxygen availability still varies in some environments. If oxygen becomes limited (a condition known as hypoxia), almost all animals react using a specific set of molecules known as the HIF pathway. This pathway – which is named after proteins called "hypoxia-inducible factors" – triggers changes that help the animal to maintain a stable level of oxygen in its cells. Yet it was not clear if the capacity to sense hypoxia and regulate oxygen demands within the body evolved in the ancestor of all animals, or if it evolved more recently.

When trying to understand early evolution, scientists often turn to some living species that sit on the oldest branches of a group's family tree. In the animal kingdom, sponges and comb jellies occupy those branches. Mills, Francis et al. have now searched the genomes of several of these animals to ask how oxygen sensing evolved. The genomes of the sponges and comb jellies surveyed lack key components of the HIF pathway, suggesting that the last common ancestor of living animals lacked the HIF pathway as well. This also implies that the ancestor of all animals probably did not respond to oxygen stress or used unknown mechanisms to deal with it instead.

In laboratory experiments, Mills, Francis et al. saw that a marine sponge named *Tethya wilhelma* does not alter its gene activity even when the oxygen levels are reduced to 0.25% of modern levels. This is consistent with the predicted absence of a HIF pathway or anything similar. Together these finding may indicate that the last common ancestor of all living animals maintained normal gene activity even at very low concentrations of oxygen.

These findings help scientists understand how life and the global environment have shaped each other since the origin of life over 3.5 billion years ago. This fundamental knowledge may provide the context needed to help society navigate through current and on-going environmental changes, including the dropping oxygen levels in the world's oceans.

DOI: https://doi.org/10.7554/eLife.31176.002

in sponges (Phylum Porifera) and comb-jellies (Phylum Ctenophora) – the two most likely candidates that have been discussed as the sister-lineage to all remaining animals (e.g. *King and Rokas, 2017*), but see *Simion et al. (2017)*; *Feuda et al., 2017*). While previous experiments on the demosponge *Halichondria panicea* demonstrated survival under 1.6–12.7 µM $O_2$ (*Mills et al., 2014*), no transcriptional data from sponges kept under low-oxygen have been reported (*Mentel et al., 2014*). Therefore, it remains unclear whether the HIF pathway extends to all living animals, which would parsimoniously suggest that HIF is an ancestral feature of animal life.

## Results

### Orthologs of HIF are absent in sponges and ctenophores

Using comparative genomics (see data sources in *Supplementary file 1*), we found that all animal groups – as well as the single-celled eukaryotic outgroup, *Capsaspora owczarzaki* (*Suga et al., 2013*) – have bHLH-PAS family proteins (*Figure 1a,b*). No members of this protein family were found in the genomes or transcriptomes of choanoflagellates, the sister lineage to animals (*King et al., 2008*; *Fairclough et al., 2013*; *Carr et al., 2017*). While one-to-one orthologs of HIFa are found in cnidarians and placozoans, sponges and ctenophores lack true one-to-one HIFa orthologs (*Figure 1a,b*). The phylogeny obtained for the bHLH-PAS family (*Figure 2*) reveals that vertebrate HIF1a arose from a series of duplications that gave rise to HIFa, SIM1, SIM2, NPAS1 and NPAS3, and that the function of the ancestral protein is unclear (*Figure 2*). The position of both sponge and ctenophore sequences in the tree suggests that sponge and ctenophore orthologs are sister to all HIF-SIM-NPAS (HSN) proteins, and that the lack of one-to-one HIFa orthologs reflects the primary absence of HIFa in these phyla, rather than a secondary loss.

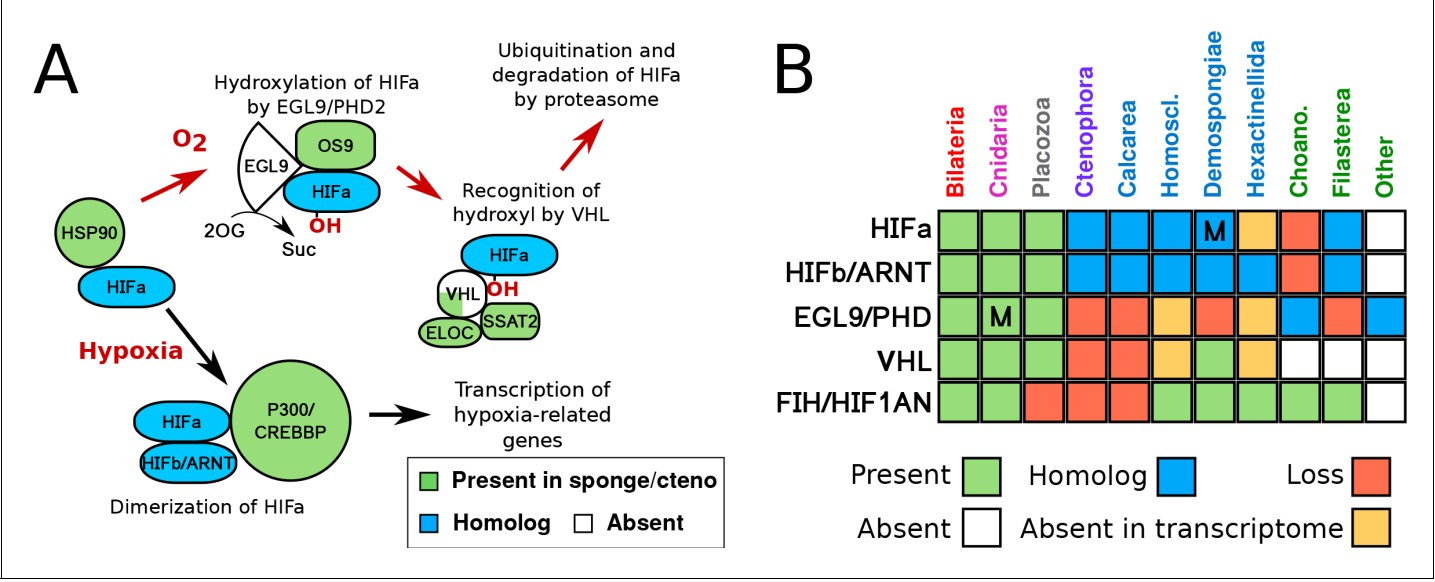

**Figure 1.** HIF pathway overview. (**A**) Schematic of HIF pathway, based on *Semenza (2007)*, showing conservation of components. Red arrows indicate the characterised oxygen-dependent pathway, which is predicted to be absent in sponges and ctenophores. (**B**) Presence-absence matrix of components of the HIF pathway across metazoans and other opisthokont groups. Labels for all four sponge classes are shown in blue. Abbreviations are: Homoscl., Homoscleromorpha; Choano., Choanoflagellata. Presence (green) refers to a 1-to-1 or 1-to-many ortholog of a protein of defined function. Homolog (blue) refers to a sister group position in trees before duplications with different or unknown functions (usually many-to-many). Secondary loss (red) refers to the gene missing in the clade, but homologs are found in non-metazoan phyla. Multiple lineage-specific duplications are indicated by the letter 'M'. Individual gene trees are shown in *Figure 2*, *Figure 2—figure supplement 1–3*.

DOI: https://doi.org/10.7554/eLife.31176.003

The following figure supplement is available for figure 1:

**Figure supplement 1.** Sulfide metabolism pathway.

DOI: https://doi.org/10.7554/eLife.31176.004

Despite the apparent absence of one-to-one orthologs in sponges, we examined whether key features of HIFa may be found in other sponge bHLH-PAS proteins. In the presence of oxygen, HIFa is hydroxylated at a key proline in the motif L[D/E]x[L/R]AP[F/Y]l that signals the protein for degradation. The motif varies between clades, where the LAPY motif in vertebrate HIFa is instead RAPY in most protostomes, RAPY or RAPF in cnidarians, and LAPF in placozoans (*Figure 3*). None of these motifs were found in any of the sequences from the 26 sponge genomes and transcriptomes in our set. However, a similar motif (LAMRAPYI) was found in the C-terminal domain of a bHLH-PAS protein from the ctenophore *Euplokamis dunlapae*, but not in any of the other eight ctenophores. The residues surrounding this motif do not align well with the ODD domains from HIFa in other animals, thus the role of this protein cannot be determined by this study. However, these results must be interpreted cautiously, particularly as much of the data for our study species derive from transcriptomes. The C-terminal domains of nearly all bHLH-PAS proteins are enriched for repeats of proline, serine, and glutamine, which makes them difficult to align and may cause downstream problems in tree generation. As prolines are highly abundant (*Figure 3b*), LAP or RAP motifs can be identified in many sponge sequences by chance, but also proteins that are not known to have oxygen-dependent degradation, including human NPAS1 and NPAS3, suggesting that this short motif may occur without functional relevance. Additionally, prolines have a unique structural role in proteins as they are mostly inflexible, meaning they rarely can be exchanged for other amino acids. For this reason, alignment programs may cluster sequences around prolines, resulting in apparent alignments of non-homologous domains (*Figure 3c,d*). Since we do not find the full hydroxylation motif in any sponge bHLH-PAS protein, none of these proteins are expected to function like HIFa.

## Key components of the HIF pathway are absent in sponges and ctenophores

VHL orthologs were not found in any ctenophore or sponges outside of a few demosponges (*Figure 1b*, *Figure 2—figure supplement 1*). We were also unable to find EGL9/PHD orthologs in any sponge or ctenophore, although we found related genes in microbial eukaryotes, including choanoflagellates (*Figure 1b*, *Figure 2—figure supplement 2*). We were, however, able to identify homologs of the Factor Inhibiting HIF (FIH/HIF1AN) in sponges (*Figure 1b*, *Figure 2—figure supplement 3*), a protein involved in regulation of HIF by hydroxylating C-terminal asparagines (*Zhang et al., 2010*). However, this protein has other targets (*Wilkins et al., 2012*) and could serve a more general function in sponges, unrelated to oxygen metabolism. Overall, the absence of essential components of the HIF pathway in sponges and ctenophores further suggests that these taxa are unable to modulate their transcriptional state in response to low oxygen levels, at least not in the same way as the remaining animal phyla.

## *Tethya wilhelma* transcriptomes remain unchanged down to oxygen levels of 0.25% atmospheric saturation

To test whether sponges regulate their transcription in response to low-oxygen availability, we examined transcriptomes from clones of the demosponge *Tethya wilhelma* (*Sarà et al., 2001*) kept at different oxygen levels over the course of 4 days. Between our low-oxygen treatments, which involved progressively lowering the oxygen concentration until it reached 0.25–2% of modern atmospheric saturation (AS) (0.5–4 $\mu$M $O_2$ at 26°C and a salinity of 32), and our high-oxygen controls (94–100% AS or 198–211 $\mu$M $O_2$), we found 128 differentially expressed genes (Benjamini-Hochberg corrected p-value<0.01) out of the 37,000 total genes predicted in the genome (*Francis et al., 2017*) (*Supplementary file 2*). None of these differentially expressed genes have predicted functions related to metabolism or stress (based on similarity to any SwissProt protein or *Amphimedon queenslandica* proteins), while 46 have no matches to any known protein in our set. This response contrasts sharply with HIF-mediated changes in other animals and supports the assessment that sponges lack the HIF pathway, and have not independently evolved a comparable oxygen-sensing transcription factor. For example, in the placozoan *Trichoplax adhaerens*, exposure to 10 $\mu$M $O_2$ resulted in the upregulation of glycolytic enzymes, and death within 5 hr (*Loenarz et al., 2011*). In cnidarians, polyps and medusae of *Aurelia* sp.1 exhibited higher expression of HIFa after exposure to 16 $\mu$M $O_2$ for 18 hr, relative to polyps and medusae kept under 250 $\mu$M $O_2$ for the same length of time (*Wang et al., 2014*). Therefore, the transcriptional response of *T. wilhelma* to low oxygen is clearly distinct from the responses seen in HIF-bearing non-bilaterian animals (i.e. placozoans and the cnidarians).

After exposing *T. wilhelma* to complete anoxia in a closed, non-circulating system for 1 hr, 5981 genes were differentially expressed (Benjamini-Hochberg corrected p-value<0.01) relative to the 4 day, high-oxygen control experiments. Of this set, 569 genes were differentially overexpressed (log fold change $\geq$ 2) and 142 genes were differentially underexpressed (log fold change $\leq$ −2) in treated vs. control sponges (*Supplementary file 3*), and include some metabolic genes (glycogen debranching enzyme, three heme binding proteins, 10 ubiquitin ligases), implying that the sponges may be stressed by these conditions. As with the low-oxygen treatment, half of the differentially expressed genes lack a reliable BLAST hit to an annotated protein, making it difficult to define which biological processes are most affected. Curiously, one of the most strongly upregulated genes is a cryptochrome, a light-dependent regulator of bHLH-PAS proteins involved in circadian rhythms (*Rivera et al., 2012*), suggesting potential for cross-talk between circadian rhythms and redox state, as seen in some vertebrates (*Rutter et al., 2001*; *Hirayama et al., 2007*). All six bHLH-PAS proteins were upregulated to some degree, whereby the log fold change was greater than one for only the HSN-like group one proteins (see *Figure 2*) and the ARNT homolog, although it is unclear what controls their expression, whether any of these proteins regulate themselves, or what the downstream targets may be. We did not explore how long *T. wilhelma* can survive complete anoxia. Overall, our data suggests that metabolic function in *T. wilhelma* is unaffected by oxygen levels as low as 0.25% AS, the lowest levels observed, and that complete anoxia in still water induces stress – at least when assayed at the mRNA level.

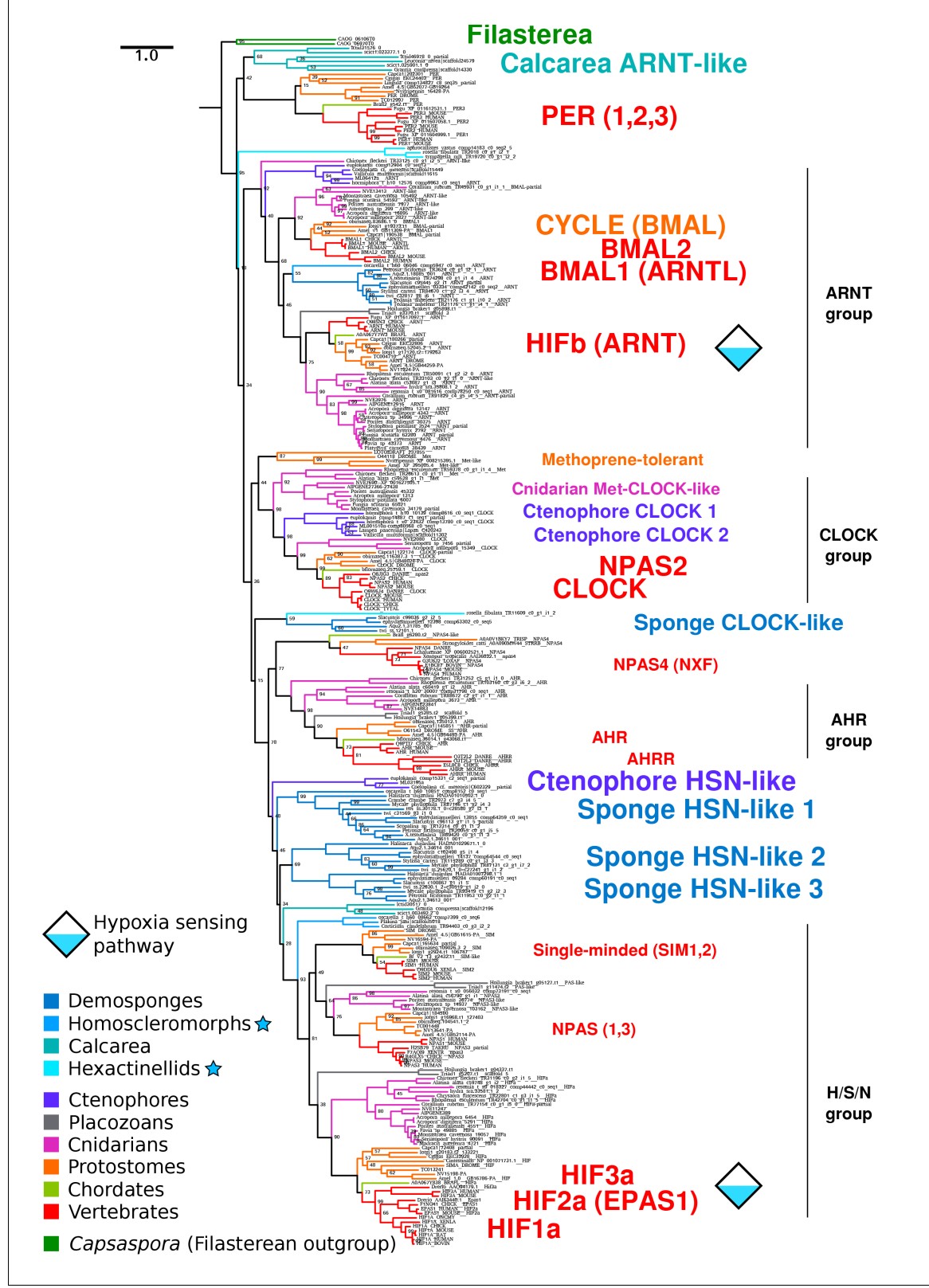

**Figure 2.** Complete bHLH-PAS tree. Phylogenetic tree of bHLH-PAS proteins across metazoa, generated with RAxML using the PROTGAMMALG model. Bootstrap values of 100 are removed for clarity. Sponge classes represented only by transcriptomes are indicated by blue stars.

DOI: https://doi.org/10.7554/eLife.31176.005

*Figure 2 continued on next page*

*Figure 2 continued*

The following figure supplements are available for figure 2:

**Figure supplement 1.** Von Hippel-Lindau protein tree.
DOI: https://doi.org/10.7554/eLife.31176.006
**Figure supplement 2.** EGL9 protein tree.
DOI: https://doi.org/10.7554/eLife.31176.007
**Figure supplement 3.** Factor-inhibiting-HIF tree.
DOI: https://doi.org/10.7554/eLife.31176.008

### *Tethya wilhelma* contracts down to oxygen levels of 4% of atmospheric saturation

*T. wilhelma* performs periodic full-body contractions under air-saturated oxygen levels (*Nickel, 2004*). Accepting this as normal behavior, we tested whether contractile dynamics were influenced by reduced oxygen concentrations. During our 4-day experiments, we observed no significant difference in contraction rate under oxygen concentrations between 4 and 100% AS (8.45–211 $\mu$M $O_2$) (*Figure 4a*, *Video 1*). Full-body contractions did cease, however, at $\leq$1.86% AS (3.9 $\mu$M $O_2$) (*Figure 4a*). We are uncertain as to why contractions ceased under these low-oxygen levels, but in separate oxygen drawdown experiments, we observed elevated rates of oxygen uptake during the contraction period (*Figure 4d*), potentially signifying that the sponges were energetically limited under $\leq$1.86% AS. However, the transcriptomes of the treatment and control sponges did not significantly differ (*Figure 5a*; Adonis Pseudo-F = 1.6138, p=0.06), showing that the cessation of contraction was not associated with any shifts in gene regulation. Furthermore, local sub-contractions across the sponge surface resumed under oxygen levels as low as 0.25% AS (0.53 $\mu$M $O_2$). In summary, independently of whether these sponges were unable to contract, or no longer needed to – perhaps to regulate their internal redox balance, as seen in the demosponge *Geodia barretti* (*Hoffmann et al., 2005*) – exposure to 0.25–1.86% AS (0.53–3.9 $\mu$M $O_2$) was not associated with any visible or transcriptional indicators of stress or death, suggesting that *T. wilhelma* remains viable under these levels.

## Discussion

### Evolution of the HIF pathway and metazoan oxygen sensing

As shown here, the HIF pathway evolved only once in animals, after the last common ancestor (LCA) of Bilateria + Cnidaria + Placozoa split from sponges and ctenophores. Therefore, the HIF pathway is not a universal metazoan trait, as previously thought (*Loenarz et al., 2011*). While sponges and ctenophores lack the capacity to sense oxygen via transcriptional regulators like HIF, they likely detect and respond to oxygen availability at the cellular level via other mechanisms, such as the allosteric regulation of heme proteins, or changes in the conductance of $O_2$-sensitive ion channels (*Chandel and Schumacker, 2000*). At the mitochondrial level, electron transport and oxidative phosphorylation cease under anoxia, while hypoxia reversibly lowers the maximum reaction rate ($V_{max}$) of the cytochrome C oxidase, thereby lowering the mitochondrial redox state and promoting the production of reactive oxygen species (*Chandel and Schumacker, 2000*). Cytochrome C is also inhibited by hydrogen sulfide (*Powell and Somero, 1986*; *Cooper and Brown, 2008*), which may serve as an indirect mechanism of oxygen detection, as environmental or metabolic sulfide can accumulate under anoxia and is removed in the mitochondria by a series of $O_2$-dependent reactions to produce sulfate. Most of the enzymes involved in this sulfide-removal pathway are found in all non-bilaterian groups (*Figure 1—figure supplement 1*), implying that these mechanisms are conserved, and that the absence of oxygen can be more universally detected among animals at the mitochondrial level through the presence of sulfide.

### bHLH-PAS transcription factors evolved only once in holozoans

Many metazoan transcription factors contain bHLH domains with diverse roles (*Simionato et al., 2007*). PAS domains are found universally, including in the oxygen-sensing FixL in bacteria (*Green and Paget, 2004*), and the circadian rhythm protein WC1 in the fungus *Neurospora crassa*

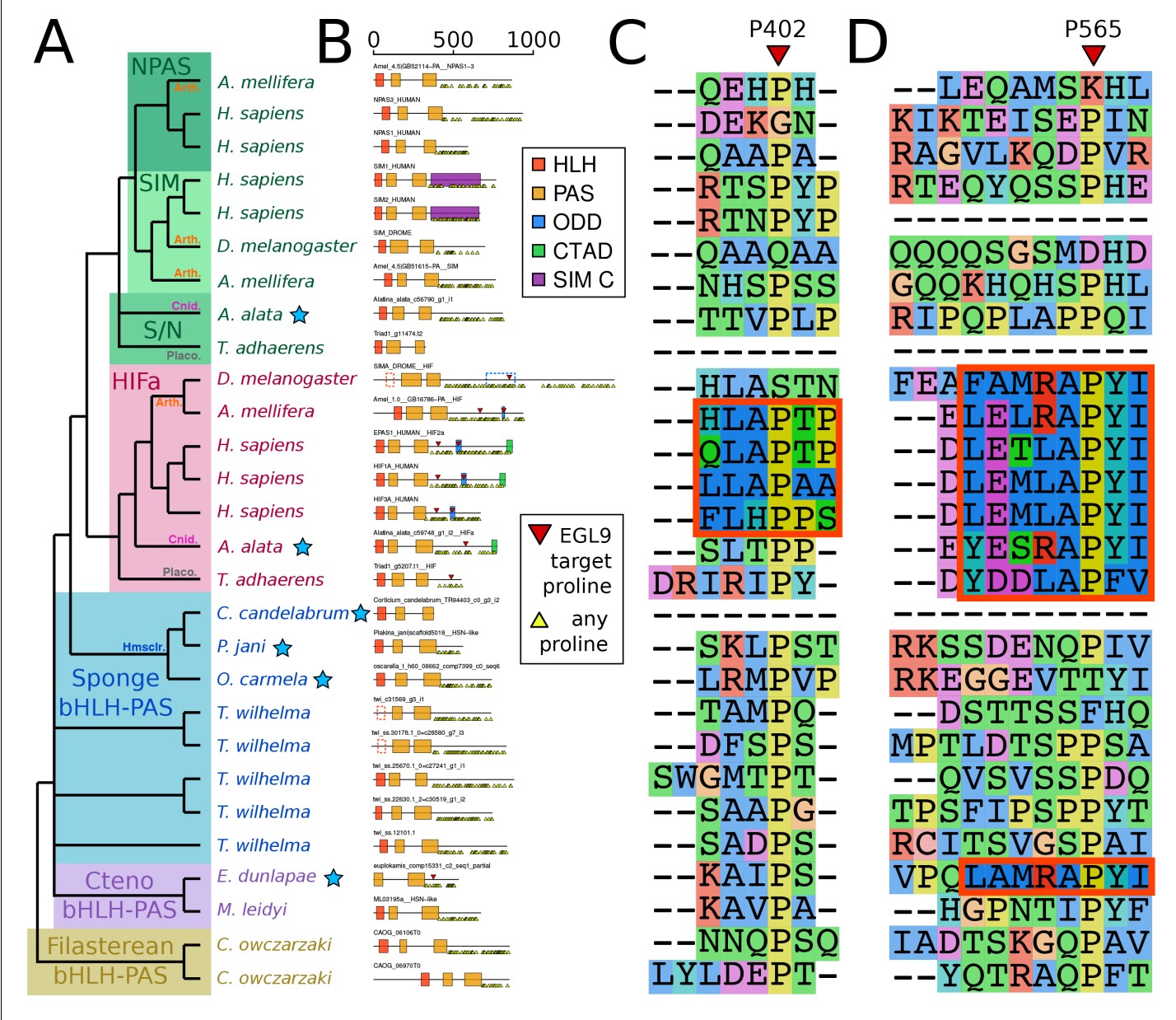

**Figure 3.** Domain organization of HIFa and related proteins. (A) Schematic tree based on *Figure 2*. Blue stars indicate the sequence was derived from a transcriptome, rather than a genome. Abbreviations for certain phyla are: Arth, arthropods; Cnid, cnidarians; Placo, placozoans; Hmsclr, homoscleromorph sponges. (B) Domain organization of the proteins, identified by hmmscan against the PFAM database. Scale bar refers to length of the protein in amino acids. Some domains were not found, although were annotated in the SwissProt entries of the canonical proteins; these are shown as dashed lines. Prolines annotated as targets of EGL9 are shown in red triangles, while all prolines in the C-terminal domain of each protein are shown below each line as yellow triangles. (C) and (D), aligned positions surrounding P402 and P565 (in human HIF1a). The matching motifs are indicated by the red boxes.

DOI: https://doi.org/10.7554/eLife.31176.009

(*Tauber et al., 2004*). However, the combination of the bHLH and PAS domains was found only in metazoans and the single-celled eukaryote *C. owczarzaki*, showing that this domain combination pre-dates the origin of animals (*Sebé-Pedrós et al., 2012*).

PAS domains have binding pockets capable of holding small molecules, such as heme in the case of fixL, and flavin for WC1. Recent crystal structures of several mouse bHLH-PAS proteins suggest potential binding pockets in both PAS domains, as well as the dimerization interface (*Wu et al.,*

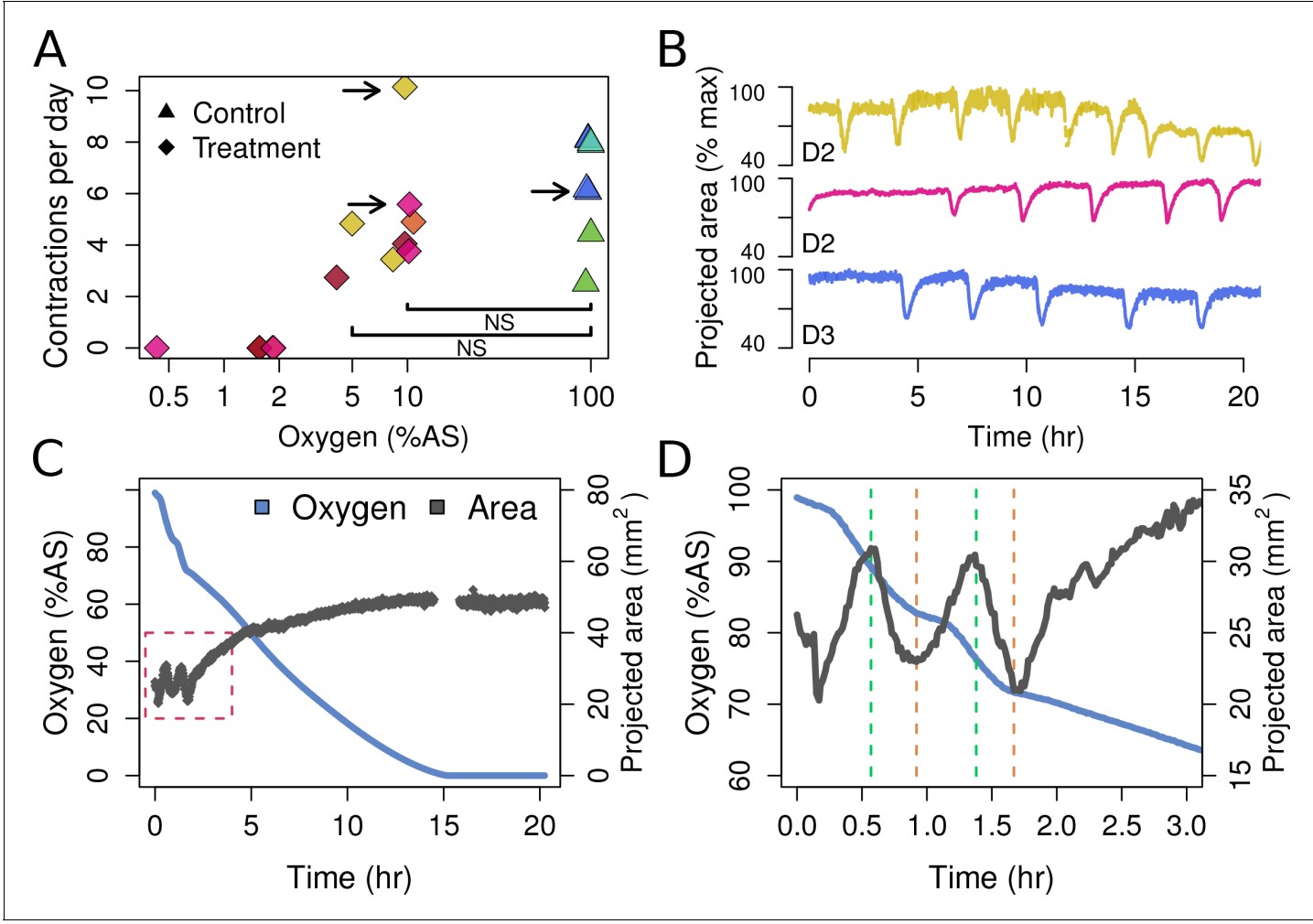

**Figure 4.** Contraction behavior. (**A**) Contraction frequency (number of contractions per day) at different stable $O_2$ levels. Each color shows a single individual across multiple treatment days; NS = not significant differences detected between the groups (i.e. 10% and 5% $O_2$ vs. all control conditions). (**B**) Contraction traces of three representative sponges (shown by arrows in A) under normal $O_2$ (blue) and hypoxia (yellow and magenta). (**C**) $O_2$ levels and projected sponge area against time within a respiration vial, highlighting the changes in $O_2$ uptake associated with sponge contraction cycles. $O_2$ was measured, and the photos were taken, every 60 s. The gap at 15 hr was due to changes in ambient light. (**D**) Expansion of the red-box from part C to highlight the increased oxygen consumption during the contraction phases (between green and red lines).

DOI: https://doi.org/10.7554/eLife.31176.010

2016). Thus, even bHLH-PAS proteins uninvolved in the HIF pathway could be indirectly affected by the redox state of ligands, such as flavins. Conceivably, the presence of oxidized or reduced factors, such as FAD or NADH, could affect dimerization, which may affect downstream transcription.

## HIF-like ODD motif is found in only one ctenophore

We identified a bHLH-PAS protein with a similar ODD motif (*Figure 3d*) in the ctenophore *E. dunlapae*, presenting a conundrum of whether HIFa was ancestrally present in ctenophores and then lost. We consider this possibility unlikely based on the bHLH-PAS tree (*Figure 2*), where the *E. dunlapae* protein groups with other ctenophore proteins and not with HIFa (or with SIM/NPAS). Since we were unable to find EGL9 or VHL in any available ctenophore transcriptome or genome, the function of the *E. dunlapae* bHLH-PAS protein is unlikely be the same as HIFa, or at least cannot involve the same binding partners. Interestingly, *E. dunlapae* belongs to the sister group to all other ctenophores (*Simion et al., 2015*), yet the homologs of the *E. dunlapae* protein in other ctenophores lack the ODD motif, raising the question of how and why the ODD motif was lost in the rest of the phylum. Overall, additional experimental investigation and genome sequencing – particular on *E.*

*dunlapae* and closely related species – are still needed to resolve the question of oxygen regulation in ctenophores.

## HIF pathway acquisition is independent of the phylogenetic position of ctenophores and is a unifying character of the clade Bilateria + Cnidaria + placozoa

The phylogenetic position of ctenophores has been a topic of recent controversy (*King and Rokas, 2017*, e.g. *Ryan et al., 2013*; *Pisani et al., 2015*). The majority of analyses recovered either sponges or ctenophores as sister to all other animals, but agreed on the position of Placozoa as the sister to a Cnidaria + Bilateria clade. Most components of the HIF pathway are absent in both sponges and ctenophores, with all the components present in the clade of Bilateria + Cnidaria + Placozoa. Therefore, the acquisition of the HIF pathway is independent of the phylogenetic position of ctenophores, as it occurred after the divergence of both ctenophores and sponges from all other animals (*Figure 6*). Furthermore, recent molecular clock estimates suggest that all non-bilaterian phyla emerged in rapid succession between 850 and 820 Ma, and that this timing was independent of the phylogenetic position of ctenophores (*Dohrmann and Wörheide, 2017*).

## Sponges and ctenophores live in low-oxygen habitats

Despite lacking the HIF pathway, both sponges and ctenophores live under low oxygen in nature. Sponges are found at oxygen concentrations as low as ~3–8 µM $O_2$ along the lower boundary of the Peruvian oxygen-minimum zone (OMZ) (*Mosch et al., 2012*), and at ~4 µM $O_2$ on the lower summit of the Volcano 7 Seamount, Eastern Tropical Pacific (*Levin, 2003*). Therefore, natural sponge populations live at oxygen concentrations approaching those under which we observed normal transcription in *T. wilhelma* (<4 µM $O_2$). Some sponges can also withstand seasonal anoxia. For example, marine sponges in Lake Lough Hyne, Ireland can apparently endure anoxia during summer thermal stratification (*Bell and Barnes, 2000*), and freshwater sponge gemmules can survive anoxia for months (*Reiswig and Miller, 1998*). Therefore, at least some sponge species can establish themselves under low-oxygen levels in nature, despite lacking the HIF pathway.

Ctenophores, the only other animal phylum predicted to lack the HIF pathway, are well known for inhabiting low-oxygen environments (*Purcell et al., 2001*). For example, in the OMZ of the North Chilean coast, the highest abundance of ctenophores (as determined through rRNA abundance) was found at oxygen levels of 6.1 µM $O_2$, defining the upper oxic-anoxic interface (*Parris et al., 2014*). Ctenophores were also found, although at lower abundance, in the anoxic core of the OMZ (*Parris et al., 2014*). Similarly, in the Eastern Tropical North Pacific, plankton nets collected the highest abundance of ctenophores in OMZ waters with oxygen concentrations as low as 3–30 µM (*Beatteay, 2012*). Overall, ctenophores have low metabolic oxygen demands, and feature most metabolically active cells in direct contact with the environment (*Thuesen et al., 2005*), thereby facilitating life under low-oxygen (*Purcell et al., 2001*). Cnidarians, which unlike ctenophores possess the HIF pathway (*Loenarz et al., 2011*; *Wang et al., 2014*), have similar distributions along modern oxygen gradients (*Purcell et al., 2001*; *Parris et al., 2014*), demonstrating that adaptation to life in low-oxygen environments is not predicated on the possession of HIF.

Despite their morphological simplicity, placozoans appear to be relatively sensitive to low-oxygen conditions, as seen in *T. adhaerens*, which dies after 5 hr of exposure to 10 µM $O_2$ (*Loenarz et al., 2011*). In nature, placozoans are found in shallow coastal waters <20 m deep (*Eitel et al., 2013*) and have not, at least to our knowledge, been reported from any low-oxygen systems. Placozoans phagocytose microbial prey

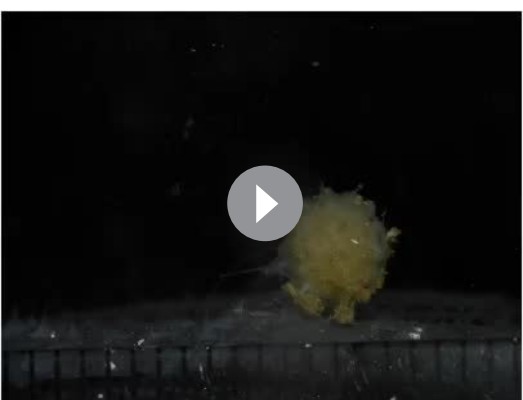

**Video 1.** Experimental video of *Tethya wilhelma* contracting at 21.1 µM $O_2$ (10% atmospheric saturation at T = 26°C and S = 32) over the course of 8 hr.
DOI: https://doi.org/10.7554/eLife.31176.012

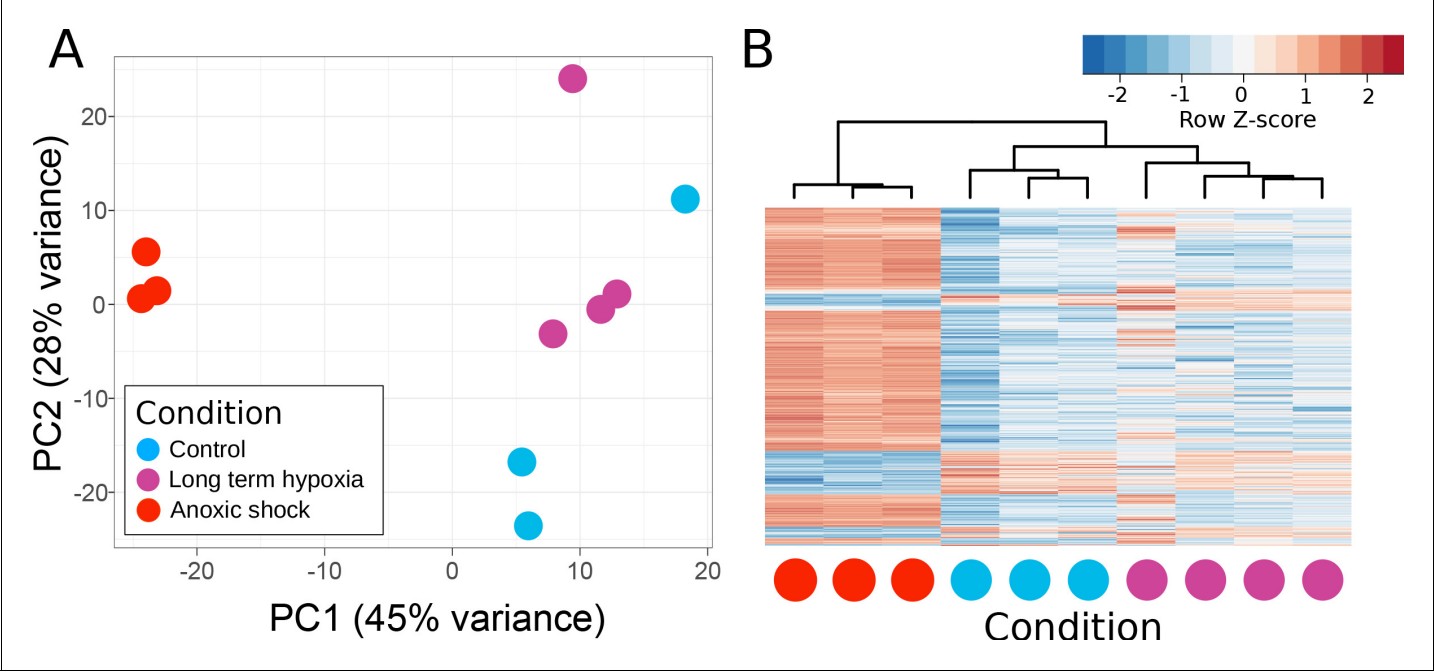

**Figure 5.** Principle components of *T. wilhelma* transcriptomes. (A) PCA plot of variance of expressed genes, where each point represents the transcriptome from a single sponge specimen at the end of the experiment. Input data were normalized read-mapping counts per gene for each sample. Samples exposed to long term hypoxia were not significantly different (Adonis Pseudo-F = 1.6138, p=0.06) from controls. (B) Heatmap showing expression of the 739 differentially expressed genes across all samples and treatments. Red and blue intensities indicate differentially overexpressed and underexpressed genes, respectively.

DOI: https://doi.org/10.7554/eLife.31176.011

via their upper epithelia, and feed through external digestion and osmotrophy via their lower epithelia (*Eitel et al., 2013*), and therefore occupy trophic levels predicted to be supported in low-oxygen environments (*Fenchel and Finlay, 1995*). Indeed, most filter-feeding sponges (in contrast to carnivorous sponges) also subsist on small microbial prey and dissolved organics, and therefore occupy the same trophic levels as placozoans (*Mills and Canfield, 2017*). The sensitivity of placozoans to low-oxygen conditions is therefore difficult to explain in terms of their body plan and trophic style.

All metazoans outside of ctenophores and sponges are predicted to have the HIF pathway (*Loenarz et al., 2011*). Outside of animals, similar mechanisms of transcriptional regulation by oxygen-dependent degradation of transcription factors have been identified in fungi (*Lee et al., 2009*), the social amoeba *Dictyostelium* (*van der Wel et al., 2005*), and plants (*Licausi et al., 2011*) – although these systems all use different protein components. Overall, it appears that oxygen-dependent transcriptional control has evolved convergently in multiple multicellular eukaryotic lineages.

## Hypoxia-dependent transcription in the last common ancestor of animals

The ancestral absence of the HIF pathway in metazoans suggests that stem-group metazoans and the metazoan LCA did not regulate their transcription in response to oxygen availability. While it is unclear how much oxygen crown-group animals ancestrally required (*Mills and Canfield, 2014*), animals most likely originated prior to the establishment of modern atmospheric oxygen levels and the permanent oxygenation of the deep ocean (*Lenton and Daines, 2017*). In this case, the ancestral absence of HIF in animals suggests that the earliest metazoans could have functioned and respired under environmental oxygen concentrations as low as 0.25% of modern atmospheric saturation, and perhaps even lower, and did not need to respond to low-oxygen availability at the transcriptional level.

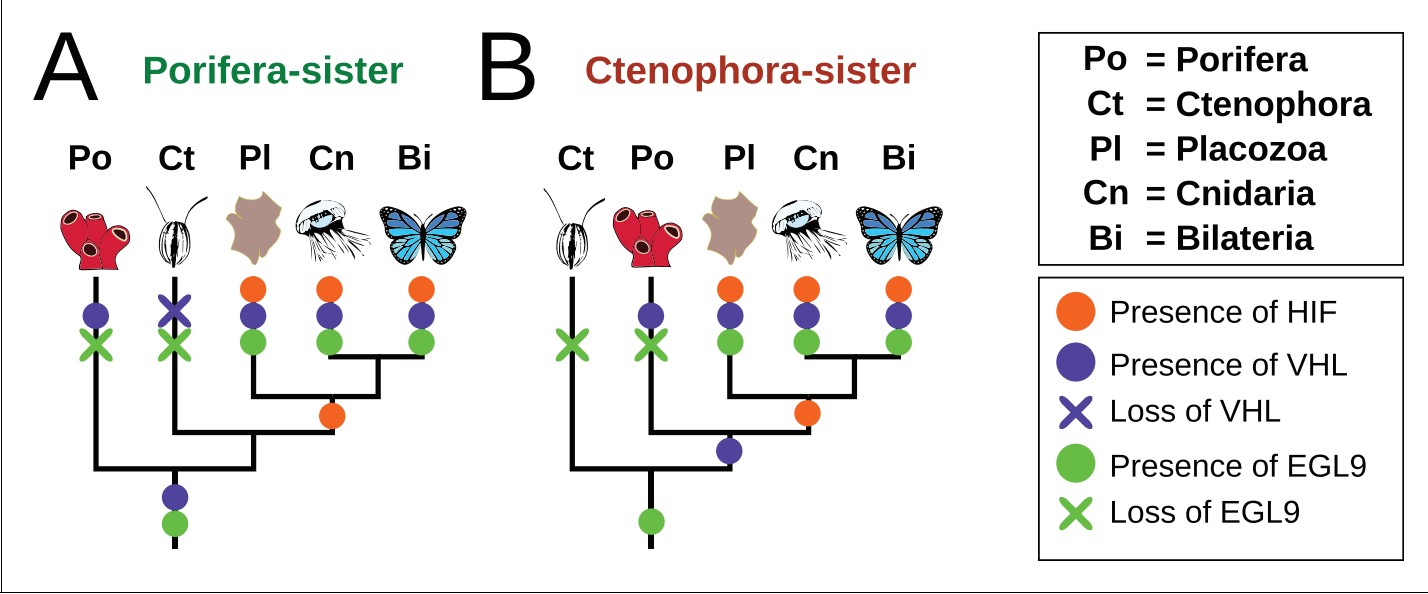

**Figure 6.** Summary schematic of the evolution and distribution of HIF pathway components in metazoans. (**A**) Presence and predicted losses in a Porifera-sister topology and (**B**) Ctenophora-sister topology.

DOI: https://doi.org/10.7554/eLife.31176.013

## Materials and methods

### Sponge material

Our *T. wilhelma* specimens developed from buds grown in the laboratory, and were ultimately clones of specimens collected from the type locality in the aquarium of the Wilhelma Zoological and Botanical Garden (Stuttgart, Germany). Sponges were maintained in aquaria of artificial seawater kept at 26°C and salinity of 32.

### Experimental apparatus for long-term low-$O_2$ exposure

For our long-term experiments, individual sponges were placed in a flow-through system that continuously subjected the sponges to circulating aquarium water sparged with a controlled air-$N_2$ mixture. For a given experimental run, an individual sponge was placed inside a 100-mL blue cap bottle, where the sponge rested upon black polypropylene mesh netting, supported by a ring of black polyurethane foam, overlying a magnetic stir bar. For photographic documentation, the outside of the blue cap bottle was covered with black felt to enhance the contrast between the sponge and its background, with an opening in the felt for visibly exposing the sponge. The lid of the blue cap bottle was modified to accommodate two hollow glass tubes for the introduction and removal of artificial seawater. The $O_2$ content of the bottle was measured using a two-channel Firesting oxygen meter (Pyroscience, Germany), with a sensor spot affixed to the inside of the bottle, and an optical fiber mounted correspondingly on the outside.

Artificial seawater circulated throughout our system within a single, self-contained loop via a peristaltic pump set at 3.0 mL min-1. First, water was drawn out of the main aquarium through silicone tubing that entered and coiled within a 2-L glass reservoir filled with filter-sterilized (0.22 µm pore size) artificial seawater. This reservoir water was sparged with an air-$N_2$ mixture set by a gas mixer, and well mixed with a magnetic stir bar to control its $O_2$ content. After the circulating aquarium water equilibrated with the sparged reservoir water (thereby reaching the desired $O_2$ concentration), it exited the reservoir and was transported (via Iso-Versinic and glass tubing) into the 100-mL blue cap bottle containing the single *T. wilhelma* specimen.

Water exited the 100-mL blue cap bottle through the lid via overpressure and was redirected back to the aquarium system, thereby completing the circulation loop. The 2-L glass sparging reservoir and the 100-mL blue cap bottle housing the sponge both sat in a water bath maintained

at ~26°C using a circulating temperature controller, monitored with the Firesting through its external temperature function and thermometer. Firesting $O_2$ and temperature measurements were taken every 30 s using the associated Profix software (Pyroscience, Germany).

The sponge, while inside the 100-mL blue cap bottle, was photographed every 30 s with a Panasonic DMC-GX7 digital camera. Photos were subsequently uploaded, and analyzed in MATLAB using an edge-detection algorithm that identified the borders of the sponge against the background of each frame, and calculated the sponge's projected area in pixels. Absolute projected area was then calculated after calibrating the scale (a transparent metric ruler placed in front of the blue cap bottle, with the sponge positioned as close to the glass side as possible) present in certain frames.

## Procedure for long-term low-$O_2$ exposure

Each experimental run (4 in total) lasted 93.6 hr (SD ± 5.5 hr) and began with the calibration of the oxygen sensor and the introduction of the sponge to the blue cap bottle, initially filled with air-saturated water ([$O_2$] = 90–100% AS = 190–211 µM $O_2$ given T = 26°C and S = 32). The gas mixture was immediately set to 90% $N_2$ and 10% air, ultimately bringing the reservoir water, and then the circulating aquarium water, to an $O_2$ concentration of ~10% AS. This concentration was met relatively quickly (15–20 min) in the sparged reservoir water, but took several hours to level out in the blue cap bottle. The set $O_2$ level (10% AS) was then maintained for an entire 24 hr after leveling out. $O_2$ was then lowered again, leveling out ≤5% AS, where it was maintained for another 15–20 hr. Targeting the same $O_2$ levels was difficult, even with identical gas mixture and peristaltic pump settings, so while two experimental runs leveled out 4–5% AS after transitioning from 10% AS, the two other runs leveled out closer to 2% AS (perhaps indicating higher levels of $O_2$ uptake in the experimental system). After this second stable $O_2$ level was reached and maintained, $O_2$ was lowered a final time to the lowest levels investigated, 0.43–1.85% AS. These levels again were maintained for another 15–20 hr before the experiments were terminated. To account for any drift in the oxygen signal, each experimental run ended with a second calibration of the oxygen sensor. We also conducted three negative controls in the same setup, where the gas mixture was set to 0% $N_2$ and 100% air, thereby subjecting the sponges to oxygen levels > 90% AS for the duration of the experiment (91.3 hr, ±SD 4.8 hr). At the end of every low-oxygen experiment and high-oxygen control, each sponge was removed from the system and frozen in liquid $N_2$ for subsequent RNA extraction (discussed below). We analyzed the contraction data in R (package lme4) using a generalized linear mixed-effects model (GLMM) with a Poisson distribution. To account for the fact that our setting required us to sample each individual multiple times, making our analysis a repeated measures one, we included each sponge as a random effect in the model. Significance of the Between and Within effects were assessed using the function Anova (package car). Post-hoc comparisons for the factors Condition, Oxygen Level (coded as Sampling Time in the model), and their interactions were tested with the function ghlt of package multcomp.

## Procedure for anoxia-exposure experiments

Three sponges were placed into a non-circulating 100-mL blue cap bottle filled with artificial seawater degassed with $N_2$ to anoxia (<0.05% AS, 105 nM $O_2$). Adding the sponges re-introduced $O_2$ to the system, reaching ca. 10% AS, with anoxia returning 50 min later. The sponges were then kept under anoxia and still water for a total of 66 min, before they were removed and then frozen in liquid $N_2$ for subsequent RNA extraction (below).

## Gene trees

For protein trees, candidate proteins were identified by reciprocal BLAST alignment using blastp or tblastn using an e-value threshold of $10^{-5}$. Datasets are listed in *Supplementary file 3*, which included data from 26 sponge species and nine ctenophore species. All BLAST searches were done using the NCBI BLAST 2.2.29 + package (*Camacho et al., 2009*). Because most functions were described for human, mouse, or fruit fly proteins, these served as the queries for all datasets. Candidate homologs were kept for analysis if they reciprocally aligned by blastp to a query protein, which was usually human. Alignments for protein sequences were created using MAFFT v7.029b, with L-INS-i parameters for accurate alignments (*Katoh and Standley, 2013*). Phylogenetic trees were generated using RAxML-HPC-PTHREADS v8.2.9 (*Stamatakis, 2014*), using the PROTGAMMALGF

model for proteins and 100 bootstrap replicates with the 'rapid bootstrap' (-f a) algorithm and a random seed of 1234.

## Domain analysis

Domains were identified in bHLH-PAS proteins using the hmmsearch program of the HMMER package (*Eddy, 2011*), with a e-value cutoff of 0.1 and bitscore threshold of 10. PFAM-A was used as the query database.

## Transcriptome sequencing and analysis of differential gene expression

Total RNA was extracted using Trizol reagent. Briefly, sponges were homogenized in Trizol and extracted using Chloroform followed by an isopropanol precipitation. The RNA extractions were quantified using a Nanodrop 1000 and quality controlled using a Bioanalyzer 2100. Strand-specific libraries were prepared using Lexogen's SENSE mRNA-Seq Library Prep Kit V2, pooled and sequenced (50 bp SE) on two lanes of an Illumina HiSeq-1500. The resulting single-end reads were mapped against the *T. wilhelma* filtered gene set with Bowtie2 (*Langmead and Salzberg, 2012*) with default parameters, allowing only single-mapping. To determine the set of differential expressed genes, the count matrix was analyzed using the R package DESeq2. Differentially expressed genes, protein alignments, and the scripts used to analyse the data are available at the project repository (https://bitbucket.org/molpalmuc/sponge-oxygen). Raw reads have been deposited at NCBI SRA under BioProject accession PRJNA380886.

## Acknowledgements

We thank P Naumann for assistance with maintaining *Tethya wilhelma*, G. Büttner for her support during laboratory work and library preparation, S Krebs and H Blum for assistance with sequencing, and A Glud for assistance with oxygen monitoring. SV thanks N Villalobos T, M Vargas V, S Vargas V and S Vargas V for their constant support. DBM was supported through grants to DEC by the Villum Foundation (Grant No. 9278), the European Research Council (Grant No. 267233), and the Danish National Research Foundation (Grant DNRF53). DBM also thanks J Glass, R Glud, and E Sperling for critically discussing an earlier version of this manuscript. DEC acknowledges funding from the Villum Foundation, through the Villum Investigator Program. GW acknowledges funding through the LMU Munich's Institutional Strategy LMUexcellent within the framework of the German Excellence Initiative (Project MODELSPONGE) and by the German Research Foundation (DFG, Grant No. Wo896/15-1).

## Additional information

### Funding

| Funder | Grant reference number | Author |
| --- | --- | --- |
| Danmarks Grundforsknings-fond | DNRF53 | Donald E Canfield |
| H2020 European Research Council | 267233 | Donald E Canfield |
| Villum Fonden | 9278 | Donald E Canfield |
| Villum Fonden | Villum Investigator Program | Donald E Canfield |
| Ludwig-Maximilans-Universität München | LMUexcellent - MODELSPONGE | Gert Wörheide |
| Deutsche Forschungsge-meinschaft | Wo896/15-1 | Gert Wörheide |

The funders had no role in study design, data collection and interpretation, or the decision to submit the work for publication.

## Author contributions
Daniel B Mills, Conceptualization, Resources, Data curation, Formal analysis, Validation, Investigation, Visualization, Methodology, Writing—original draft, Project administration, Writing—review and editing; Warren R Francis, Conceptualization, Resources, Data curation, Software, Formal analysis, Validation, Investigation, Visualization, Methodology, Writing—original draft, Writing—review and editing; Sergio Vargas, Conceptualization, Resources, Data curation, Formal analysis, Validation, Investigation, Methodology, Writing—review and editing; Morten Larsen, Resources, Methodology, Writing—review and editing; Coen PH Elemans, Resources, Software, Writing—review and editing; Donald E Canfield, Conceptualization, Resources, Supervision, Funding acquisition, Project administration, Writing—review and editing; Gert Wörheide, Conceptualization, Resources, Supervision, Funding acquisition, Validation, Project administration, Writing—review and editing

## Author ORCIDs
Daniel B Mills https://orcid.org/0000-0002-2565-6933
Warren R Francis http://orcid.org/0000-0003-3473-4726
Sergio Vargas http://orcid.org/0000-0001-8704-1339
Morten Larsen http://orcid.org/0000-0002-5579-0252
Coen PH Elemans https://orcid.org/0000-0001-6306-5715
Gert Wörheide http://orcid.org/0000-0002-6380-7421

## Decision letter and Author response
Decision letter https://doi.org/10.7554/eLife.31176.023
Author response https://doi.org/10.7554/eLife.31176.024

# Additional files

## Supplementary files
• Supplementary file 1. Genomes and transcriptomes used for searches and gene trees
DOI: https://doi.org/10.7554/eLife.31176.014

• Supplementary file 2. Differentially expressed genes following the long-term hypoxia experiment
DOI: https://doi.org/10.7554/eLife.31176.015

• Supplementary file 3. Differentially expressed genes following the shock experiment
DOI: https://doi.org/10.7554/eLife.31176.016

• Transparent reporting form
DOI: https://doi.org/10.7554/eLife.31176.017

## Major datasets
The following datasets were generated:

| Author(s) | Year | Dataset title | Dataset URL | Database, license, and accessibility information |
| --- | --- | --- | --- | --- |
| Francis WR, Mills DB, Vargas S, Wörheide G | 2017 | Data for paper | https://bitbucket.org/molpalmuc/sponge-oxy-gen | Publicly available at Bitbucket (https://bitbucket.org/) |
| Vargas S, Francis WR, Wörheide G | 2017 | Raw sequence data | https://www.ncbi.nlm.nih.gov/bioproject/380886 | Publicly available at the NCBI BioProject (accession no. PRJNA380886) |

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
