## [Decision Letter]

Thank you for submitting your article "Late acquisition of the hypoxia-inducible factor in animals" for consideration by *eLife*. Your article has been reviewed by three peer reviewers, one of whom is a member of our Board of Reviewing Editors and the evaluation has been overseen by Ian Baldwin as the Senior Editor. The following individuals involved in review of your submission have agreed to reveal their identity: Kalle Rytkonen (Reviewer #3).

The reviewers have discussed the reviews with one another and the Reviewing Editor has drafted this decision to help you prepare a revised submission.

Summary:

In this manuscript Mills et al. report the late acquisition of hypoxia-inducible factor (HIF) in animals, as the authors find that sponges and ctenophores seem to not have the essential components of the HIF pathway encoded in their genomes (assessed by comparative genomics). The authors show convincingly that the demosponge T. wilhelmina can survive under low-oxygen conditions (0.25 -2%) and find no evidence that their transcriptional state is changed (e.g. no comparable oxygen-sensing transcription factor found upregulated when they examined transcriptomes from sponge clones kept at different oxygen levels). This current manuscript builds on previous work by some of the authors (Mills et al., 2014). There, the response of sponges to hypoxia has been investigated. In the 2014 paper it was shown that demosponges (in this case H. panacea) can survive under low-oxygen conditions of 0.5-4.0% present atmospheric levels (and very similar conditions are reported in this manuscript). In addition, a previous study by Loenarz et al., 2011, found that the hypoxia-inducible transcription factor pathway regulates oxygen sensing in another non-bilaterian animal, the placozoan T. adhaerens. The current manuscript by Mills et al., adds more to the question if one of the first animals, the sponges, were able to tolerate/live under low oxygen conditions and their findings point towards sponges being able to metabolize aerobically under very low environmental oxygen concentrations – an interesting and important finding. There are no experimental data presented for ctenophores to test for survival/change in expression level under low oxygen conditions. While the comparative genomics analysis points towards the absence of the HIF pathway in ctenophores, without experimental evidence it cannot be conclusively be ruled out that some components of the HIF pathway will be switched on under low oxygen conditions. This, together with the current dispute of the phylogenetic position of ctenophores (from sister group to animals to sister group to bilaterians) could easily change the timing of the acquisition of the HIF pathway from late to early and thus change the major message of the paper. Nevertheless, the current work is an important step and considerably advances our understanding of the evolution of the HIF pathway in animals.

Essential revisions:

1) The authors have constructed their phylogenetic trees using FastTree, a program that is not designed to produce accurate trees. This is something that is acknowledged by the authors of the software as well as by recent analyses (e.g., http://www.biorxiv.org/content/early/2017/05/25/142323). As a lot of the authors' findings are based on phylogenetic reconstruction, we strongly recommend that the authors reconstruct their gene trees using either RAxML or IQTREE, which are much more accurate, while still relatively fast. In addition, it would be more convincing if in addition to ML also Bayesian method (for example using CAT model) would be used for the analysis.

2) We found the figures of the manuscript of poor quality and not very informative. Specifically:

- The taxon names in Figure 2, Figure 3, and 4 are not readable. These 3 figures should be rendered readable and placed in the supplement. In their place, the authors should construct a figure similar to that shown in Figure 4 of by Loenarz et al., 2011.

- The meaning of Figure 1 was unclear – please explain.

- Figure 5 should be placed in the supplement.

3) The title of the paper is ambiguous. What does "late acquisition" mean? We propose the authors should revise it to something like: "The Urmetazoan lacked the HIF pathway and was able to survive in low oxygen environments".

4) Please cite your previous work (2014) in the Introduction of this manuscript and clarify the novel aspects of this new manuscript relative to the 2014 one.

5) Results section: It is somewhat unclear how exactly the comparative analysis was performed. Very little information is provided in the Materials and methods section. It is also unclear which ctenophore genomes have been searched. There are also many more ctenophore transcriptomes available. Have these been searched for HIF pathway genes? If not, they should be, and their results reported.

6) Results section: The authors claim that all sponges and all ctenophores have no one-to-one HIF1 ortholog, but instead find HIF1 homologs in these two lineages (marked blue in Figure 1 and based on a phylogenetic tree). It would very important and informative to analyze these HIF1 homologs further and show their domain architecture and their key, functionally important residues (if present) in comparison to well characterized HIF1 from other animals. Do the sponge and ctenophore HIF1 homologs (as the authors refer to it) possess NLS, BHLH, PAS ad ODDD domains?

7) Results section: The prolyl hydroxylation site motif xALP[F/Y]x of HIFa (ODD) seems to have a typing error, previously its was minimally defined as LxxLAP, that is, LAP, not ALP. Also, in Trichoplax adhaerens HIF, only LAP was conserved (Loenarz et al., 2011), and sufficient for oxygen dependent hydroxylation by PHD. Then, reference "see methods for data access" is not very practical for a reader who would just want to see protein alignments specifically for the motif. A supplementary figure showing protein alignments HIF C-terminal portions in the exemplary species (e.g. those that are show in Figure 1) and showing the presence of the motif in other species including T. adhaerens versus absence of it in sponges and Ctenophores would be useful.

8) Figure 1: Does the absent in transcriptome (yellow) mean that there is no sequenced genome available so transcriptome data is used? Please mark in the figure and write in the legend which species have their genome sequenced and for which there is only transcriptome data, and in Materials and methods section give references for these.

9) Results section to Figure 1: Calcarea EGL9 is red, but in Figure 3 there is some Calcareous sponges homologs. Should that red be blue in Figure 1, or how do you consider that long branch in Figure 3?

10) Results section: Write the definition or cut-off for differentially expressed genes in the text.

11) Supplementary file 1: Check the file, in text "49 differentially expressed genes […] 10 have no matches", but in that excel sheet there seems to be only 28 genes of which 17 have no matches. In Supplementary file 2 the numbers seem to match the text.

12) In Supplementary file 1 the used cut-off seems to be log2FoldChange > 2, and looking at the genes, the highest padj = 9.42E-05. In this this case, when absence of differential transcription is used in main argumentation, these cut-offs are quite stringent. It would be more transparent to present a list with a looser cut-off (for example fc >2 or fc > 1.5, FDR < 0.01), and then possibly discuss gene numbers with a more stringent cut-off if that still appears more suitable. Speculatively, with a looser cut-off, there might be enough genes that some category would be enriched in GO functional enrichment analysis. Showing and discussing a possible modest low-oxygen response does not need to contradict with the main argument – that is, the absence of major HIF dependent transcriptional response.

13) Discussion section: Please also shortly discuss the sponge data in relation to the Placozoan T. adherens, the simplest animal known to have HIF system, that is, what are the physiological and ecological differences between sponges+ctenophores versus Placozoa?

14) The sixth paragraph of the Discussion section says: "[…]metazoan LCA was most likely adapted to low and variable oxygen levels. " – is this contradictory to your general argument as just in previous sentence you write "trophic levels supported by both anoxic and low-oxygen systems"? Generally speaking, HIF system in other animals allowed more sophisticated fine-tuning to variable oxygen levels and is considered as a major adaptation to variable oxygen levels.

15) Subsection “Gene trees”: There is no information of the used genome datasets, please describe and provide references for the genome and transcriptome data.

[Editors' note: further revisions were requested prior to acceptance, as described below.]

Thank you for resubmitting your work entitled "The last common ancestor of animals lacked the HIF pathway and respired in low-oxygen environments" for further consideration at *eLife*. Your revised article has been favorably evaluated by Ian Baldwin (Senior editor) and three reviewers, one of whom is a member of our Board of Reviewing Editors.

The manuscript has been greatly improved and all major concerns have been satisfactorily addressed. There are, however, a few remaining issues that need to be addressed before acceptance, as outlined below:

1) Introduction, where the authors state: "the two most likely candidates for the sister-lineage to all remaining animals (Simion et al., 2017)", it would be good if they cited the several papers in favor of the one or the other alternative, rather than one of their recent papers. In addition to the Simion paper, there are several 2017 papers that should be cited here, including:

https://www.ncbi.nlm.nih.gov/pubmed/28993654, https://www.ncbi.nlm.nih.gov/pubmed/29017048, https://www.ncbi.nlm.nih.gov/pubmed/28812610, https://www.ncbi.nlm.nih.gov/pubmed/28812701, https://www.ncbi.nlm.nih.gov/pubmed/29199080

2) Figure 1 and Figure 2—figure supplement 3: HIF1N and HIF-inhibitor names are not exact. Exact names are HIF1AN (Hypoxia Inducible Factor 1 Α Subunit Inhibitor) that is better known as Factor Inhibiting HIF1 (FIH-1 or FIH). Please use one of the two established names.

3) Figure 1—figure supplement 1. Sulfide metabolism pathway."… the naming of the supplementary figures is very confusing.

4) Subsection “Orthologs of HIF are absent in sponges and ctenophores”: "RAPF in cnidarians", but in Figure 3. alata (Alatina alata) is RAPY?

5) Figure 3: What are the blue stars? Please, mark the phylogenetic clades/groups, for example that A. alata (Alatina alata) is Cnidarian.

6) Subsection “Orthologs of HIF are absent in sponges and ctenophores” and Figure 4: Authors should write how many sponge HIF homologs and how many ctenophore HIF homologs were analyzed to clarify the power of the analyses. In Figure 4, The E. dunlapae LAMRAPYI motif was predicted as EGL9 target, however in that homologue no HLH domain was predicted. Authors' argument that E. dunlapae LAMRAPYI motif could be random instead of natural selection is quite justifiable, but it might be good to clearly write that for ctenophores the whole case is less settled compared to sponges, because only 2 ctenophore HIF homologues were analyzed.

7) Subsection “Key components of the HIF pathway are absent in sponges and ctenophores”: Authors should mention that HIF1N (that is FIH) is present in some sponges. Authors should mention FIH in discussion, and point out that it has been reported to have other targets than HIF (Zhang et al., 2010, Wilkins et al., 2012, Scholz et al., 2016).

8) Subsection “Tethya wilhelma transcriptomes remain unchanged down to oxygen levels of 0.25% atmospheric saturation”: In previous essential revision 12, reviewers indirectly suggested GO functional enrichment analysis for the gene sets – this was not done. An additional supplementary table with top GO enrichment terms would give a concise presentation compared to scanning trough the list of individual gene annotations. At least for the anoxia group, few top GO terms could be mentioned in the Results section.

9) Subsection “HIF-like ODD motif is found in only one ctenophore”: The bigger question is that if there is the possibility that ctenophores ancestor had functional HIF that then was pseudogenisized? Authors could restate that based on the phylogeny the E. dunlapae homolog with ODD motif is not a HIF ortholog, and considering all the results this possibility is unlikely.

10) Subsection “HIF acquisition is independent of the phylogenetic position of ctenophores and is a unifying character of the clade Bilateria+Cnidaria+Placozoa”: In the title would you rather want to say HIF pathway acquisition rather than HIF acquisition?

11) Subsection “HIF acquisition is independent of the phylogenetic position of ctenophores and is a unifying character of the clade Bilateria+Cnidaria+Placozoa”: This could be deleted - “Since ctenophores and sponges lack HIF, we propose the name “hifozoa” for the clade Bilateria+Cnidaria+Placozoa. This clade was previously called “parahoxozoa” (Ryan et al., 2010), though the name was invalidated after finding a ParaHox gene -- the proposed uniting feature of the clade -- in the genome of the calcareous sponge Sycon ciliatum (Fortunato et al., 2014).”

12) Subsection “Hypoxia-sensing transcription factors evolved independently in other multicellular eukaryotes”: It is not very sensible to give a discussion title to a subject that is very loosely related to your study. Removing or merging this section is suggested.

---

## [Author Response]

Essential revisions:1) The authors have constructed their phylogenetic trees using FastTree, a program that is not designed to produce accurate trees. This is something that is acknowledged by the authors of the software as well as by recent analyses (e.g., http://www.biorxiv.org/content/early/2017/05/25/142323). As a lot of the authors' findings are based on phylogenetic reconstruction, we strongly recommend that the authors reconstruct their gene trees using either RAxML or IQTREE, which are much more accurate, while still relatively fast. In addition, it would be more convincing if in addition to ML also Bayesian method (for example using CAT model) would be used for the analysis.FastTree was used only for the schematic figure to give an approximate sense of branch lengths, compared to the main HIF tree, which was produced with RAxML. Thus, all trees were made with RAxML, as stated in the Materials and methods section and captions for each individual figure. Other reviewer comments indicate confusion by this tree, so we have removed it and only refer to the complete trees made with RAxML.2) We found the figures of the manuscript of poor quality and not very informative. Specifically:- The taxon names in Figure 2, Figure 3, and 4 are not readable. These 3 figures should be rendered readable and placed in the supplement. In their place, the authors should construct a figure similar to that shown in Figure 4 of by Loenarz et al., 2011.

This figure showing the alignment and domain organization has been added.

- The meaning of Figure 1 was unclear – please explain.

This panel has been removed.

- Figure 5 should be placed in the supplement.

This figure has been removed.

3) The title of the paper is ambiguous. What does "late acquisition" mean? We propose the authors should revise it to something like: "The Urmetazoan lacked the HIF pathway and was able to survive in low oxygen environments"

We have changed our original title, which was admittedly ambiguous, to “The last common ancestor of animals lacked the HIF pathway and respired in low oxygen environments.”

4) Please cite your previous work (2014) in the Introduction of this manuscript and clarify the novel aspects of this new manuscript relative to the 2014 one.

We have added this citation in the introduction and emphasized the novel aspects of the current study.

5) Results section: It is somewhat unclear how exactly the comparative analysis was performed. Very little information is provided in the Material and methods section. It is also unclear which ctenophore genomes have been searched. There are also many more ctenophore transcriptomes available. Have these been searched for HIF pathway genes? If not, they should be, and their results reported.

Both genomes and the public transcriptome sets were examined.

6) Results section: The authors claim that all sponges and all ctenophores have no one-to-one HIF1 ortholog, but instead find HIF1 homologs in these two lineages (marked blue in Figure 1 and based on a phylogenetic tree). It would very important and informative to analyze these HIF1 homologs further and show their domain architecture and their key, functionally important residues (if present) in comparison to well characterized HIF1 from other animals. Do the sponge and ctenophore HIF1 homologs (as the authors refer to it) possess NLS, BHLH, PAS ad ODDD domains?

The bHLH and PAS domains are present and easily identified in proteins. Given the ubiquity of bHLH-PAS domain proteins in metazoans, it is most parsimonious to assume that this domain combination occurred only once and thus all bHLH-PAS proteins are homologs, that is, sharing a common ancestor. We have added a paragraph to the discussion to explain this. Sponges have proteins with this domain combination, and thus are homologs. However, when examined across the entire protein family, the C-terminal domains (the CTAD or ODDD) are extremely difficult to identify because this region of most of the proteins is enriched with repeats of S, Q, and P, meaning they usually do not align correctly between animal clades (vertebrates vs cnidarians) or between genes (HIF vs SIM).

7) Results section: The prolyl hydroxylation site motif xALP[F/Y]x of HIFa (ODD) seems to have a typing error, previously its was minimally defined as LxxLAP, that is, LAP, not ALP. Also, in Trichoplax adhaerens HIF, only LAP was conserved (Loenarz et al., 2011), and sufficient for oxygen dependent hydroxylation by PHD. Then, reference "see methods for data access" is not very practical for a reader who would just want to see protein alignments specifically for the motif. A supplementary figure showing protein alignments HIF C-terminal portions in the exemplary species (e.g. those that are show in Figure 1) and showing the presence of the motif in other species including T. adhaerens versus absence of it in sponges and Ctenophores would be useful.

Indeed, this was a typo. We have corrected this in the text, and expanded on the explanation. When examining our complete set, it is clear that the motif is roughly Lxx[L/R]AP[F/Y]I[P/D]. We have generated another figure to show this part of the alignment on a subset of the sequences used. Our original searches did not find the LAPY motif in any ctenophore, however, as some species have RAPY instead, we actually were able to find RAPY in one species of ctenophore. The implications of this are now examined in the Discussion section, and we have added the aligned portions to a new figure. It must still be noted that EGL9 and VHL were not found in the transcriptome of that ctenophore.

8) Figure 1: Does the absent in transcriptome (yellow) mean that there is no sequenced genome available so transcriptome data is used? Please mark in the figure and write in the legend which species have their genome sequenced and for which there is only transcriptome data, and in Materials and methods section give references for these.

Absent in transcriptome means that absence could be due to coverage limits or tissue specific expression, thus cannot be assumed to mean secondary loss. Very few species have genomes available, so most sequences are from transcriptomes. The list is long and cannot really be placed into a figure, so this information is now included in an additional Table.

9) Results section: Calcarea EGL9 is red, but in Figure 3 there is some Calcareous sponges homologs. Should that red be blue in Figure 1, or how do you consider that long branch in Figure 3?

The calcarean sequences were meant to show that proteins of the same superfamily can still be identified in calcareous sponges and choanoflagellates, whereas nothing at all can be identified by BLAST in other sponges or ctenophores. This is clearly a different gene, but the function of these proteins is unknown. These sequences aside, there is a single origin of EGL9 in metazoans based on sequences from other single-celled eukaryotes, meaning the absence of EGL9 in ctenophores and all sponges is secondary loss. To avoid this confusion, we have removed these sequences from the alignment and remade the tree figure (though the original will be still available on the bitbucket repo).

10) Results section: Write the definition or cut-off for differentially expressed genes in the text.

A definition is now provided in the text.

11) Supplementary file 1: Check the file, in text "49 differentially expressed genes […]. 10 have no matches", but in that excel sheet there seems to be only 28 genes of which 17 have no matches. In Supplementary file 2 the numbers seem to match the text.

We checked the DESeq2 output again and there are 28 DEGs as in the table. The number 49 was a mistake in the previous version of manuscript. We have corrected the manuscript to address the next comment by the reviewers and now provide different numbers of DEGs.

12) In Supplementary file 1 the used cut-off seems to be log2FoldChange > 2, and looking at the genes, the highest padj = 9.42E-05. In this this case, when absence of differential transcription is used in main argumentation, these cut-offs are quite stringent. It would be more transparent to present a list with a looser cut-off (for example fc >2 or fc > 1.5, FDR < 0.01), and then possibly discuss gene numbers with a more stringent cut-off if that still appears more suitable. Speculatively, with a looser cut-off, there might be enough genes that some category would be enriched in GO functional enrichment analysis. Showing and discussing a possible modest low-oxygen response does not need to contradict with the main argument – that is, the absence of major HIF dependent transcriptional response.

We repeated the DESeq2 analyses and now provided the total number of differentially expressed genes found without filtering by log fold change. This is all those genes with a Benjamini-Hochberg adjusted p-value < 0.01. The total number of differentially expressed genes in our long-term treatment was 128. We have annotated this set of genes and find none of them clearly involved in stress response or metabolism. In contrast, our anoxic shock experiment resulted in 5981 differentially expressed genes. We now provide the annotated list of all differentially expressed genes for both experiments, in these tables the log fold change and adjusted p-values can be also found.

13) Discussion section: Please also shortly discuss the sponge data in relation to the Placozoan T. adherens, the simplest animal known to have HIF system, that is, what are the physiological and ecological differences between sponges+ctenophores versus Placozoa?

We now compare sponge and placozoan ecology and oxygen sensitivity in a new paragraph in the Discussion section.

14) The sixth paragraph of the Discussion section says: "[…]metazoan LCA was most likely adapted to low and variable oxygen levels. " – is this contradictory to your general argument as just in previous sentence you write "trophic levels supported by both anoxic and low-oxygen systems"? Generally speaking, HIF system in other animals allowed more sophisticated fine-tuning to variable oxygen levels and is considered as a major adaptation to variable oxygen levels.

We have removed both of these lines. However, the line about trophic levels was a reference to the idea (as argued in Mills and Canfield, 2017) that the metazoan last common ancestor primarily fed on bacteria and dissolved organic carbon, which is a trophic lifestyle supported in both low-oxygen and anoxic systems.

15) Subsection “Gene trees”: There is no information of the used genome datasets, please describe and provide references for the genome and transcriptome data.

As above, these data have been added into a supplemental Table.

[Editors' note: further revisions were requested prior to acceptance, as described below.]1). Introduction, where the authors state: "the two most likely candidates for the sister-lineage to all remaining animals (Simion et al., 2017)", it would be good if they cited the several papers in favor of the one or the other alternative, rather than one of their recent papers. In addition to the Simion paper, there are several 2017 papers that should be cited here, including:https://www.ncbi.nlm.nih.gov/pubmed/28993654, https://www.ncbi.nlm.nih.gov/pubmed/29017048, https://www.ncbi.nlm.nih.gov/pubmed/28812610, https://www.ncbi.nlm.nih.gov/pubmed/28812701, https://www.ncbi.nlm.nih.gov/pubmed/29199080

We have added more citations. However, we refrained from citing some of the suggested 2017 papers because we have recently shown (Feuda et al.,) that some of the methods applied in those studies (e.g. Shen et al., 2017, Whelan et al., 2017) were not fully adequate. However, we have cited the also suggested recent review by King and Rokas, (2017) instead, and modified the citation to read now: (e.g. King and Rokas, 2017, but see Simion et al. 2017; Feuda et al., 2017).

2) Figure 1 and Figure 2—figure supplement 3: HIF1N and HIF-inhibitor names are not exact. Exact names are HIF1AN (Hypoxia Inducible Factor 1 Α Subunit Inhibitor) that is better known as Factor Inhibiting HIF1 (FIH-1 or FIH). Please use one of the two established names.

Agreed, these have been changed in the figures.

3) Figure 1—figure supplement 1. Sulfide metabolism pathway." …the naming of the supplementary figures is very confusing.

Thank you, we have corrected the figure and figure supplement labels and references.

4) Subsection “Orthologs of HIF are absent in sponges and ctenophores”: "RAPF in cnidarians", but in Figure 3. alata (Alatina alata) is RAPY?

Upon looking back at the alignment, cnidarians have both RAPF and RAPY. This has been clarified in the text.

5) Figure 3: What are the blue stars? Please, mark the phylogenetic clades/groups, for example that A. alata (Alatina alata) is Cnidarian.

Blue stars indicate transcriptomes. This has been changed in the caption. Small clade names have also been added where it was not obvious.

6) Subsection “Orthologs of HIF are absent in sponges and ctenophores” and Figure 4: Authors should write how many sponge HIF homologs and how many ctenophore HIF homologs were analyzed to clarify the power of the analyses. In Figure 4, The E. dunlapae LAMRAPYI motif was predicted as EGL9 target, however in that homologue no HLH domain was predicted. Authors' argument that E. dunlapae LAMRAPYI motif could be random instead of natural selection is quite justifiable, but it might be good to clearly write that for ctenophores the whole case is less settled compared to sponges, because only 2 ctenophore HIF homologues were analyzed.

The reviewer raises several important and complex points. We have changed the text in the Results and Materials and methods sections to include the numbers: 26 sponges and 9 ctenophores were analysed in total. As almost all derive from transcriptomes, many were too fragmented to be used at all and were therefore excluded from further analyses. Thus, only 3 ctenophore HSN-like proteins were in the final alignment. The lack of the HLH domain in E. dunlapaeis likely due to low coverage of this transcript, which can lead to assembly problems, though most of the rest of the transcript was present, so we used it. We are still puzzled by this motif, as it is unusually similar to the HIF motifs, yet the rest of the evidence (absence of EGL9/VHL) suggests it may not function this way. We agree that the situation is unclear in ctenophores, and have added a sentence in the discussion highlighting this.

7) Subsection “Key components of the HIF pathway are absent in sponges and ctenophores”: Authors should mention that HIF1N (that is FIH) is present in some sponges. Authors should mention FIH in discussion, and point out that it has been reported to have other targets than HIF (Zhang et al., 2010, Wilkins et al., 2012, Scholz et al., 2016).

Good point, this has been changed in the text.

8) Subsection “Tethya wilhelma transcriptomes remain unchanged down to oxygen levels of 0.25% atmospheric saturation”: In previous essential revision 12, reviewers indirectly suggested GO functional enrichment analysis for the gene sets – this was not done. An additional supplementary table with top GO enrichment terms would give a concise presentation compared to scanning trough the list of individual gene annotations. At least for the anoxia group, few top GO terms could be mentioned in the Results section.

We have now done the suggested analysis to retrieve GO terms. Indeed, many proteins matched some domains (approx 70% in the hypoxia set, 50% in the shock set), but as many domains lack functional annotations (approx 1/3 of those with detectable domains), over half of the proteins in both sets do not have predicted functions or GO annotations. In addition, the majority of the predicted processes cannot be taken into account by existing annotation tools, as the top GO terms tend to be too broad (a caveat described on the gene ontology website). For instance, the top predicted functions are “protein binding”, “ATP binding” and “GTP binding”. Because of this, we think that any aggregate analysis risks being uninformative or speculative and prefer to exclude this information from the main text. The tables showing these results have been placed on the online bitbucket repo with the rest of the analysis files in case the reader wants to see these results. Despite the difficulties in providing an analysis of the GO terms, and in the interest of providing some depth to the biological analysis of these genes, we have added several sentences about the putative function of a few of the most upregulated genes with the highest confidence hits.

9) Subsection “HIF-like ODD motif is found in only one ctenophore”): The bigger question is that if there is the possibility that ctenophores ancestor had functional HIF that then was pseudogenisized? Authors could restate that based on the phylogeny the E. dunlapae homolog with ODD motif is not a HIF ortholog, and considering all the results this possibility is unlikely.

This is related to point 6 above. We have added a sentence to the Discussion section.

10) Subsection “HIF acquisition is independent of the phylogenetic position of ctenophores and is a unifying character of the clade Bilateria+Cnidaria+Placozoa”: In the title would you rather want to say HIF pathway acquisition rather than HIF acquisition?

Indeed, we have changed the title accordingly.

11) Subsection “HIF acquisition is independent of the phylogenetic position of ctenophores and is a unifying character of the clade Bilateria+Cnidaria+Placozoa”: This could be deleted “Since ctenophores and sponges lack HIF, we propose the name “hifozoa” for the clade Bilateria+Cnidaria+Placozoa. This clade was previously called “parahoxozoa” (Ryan et al., 2010), though the name was invalidated after finding a ParaHox gene -- the proposed uniting feature of the clade -- in the genome of the calcareous sponge Sycon ciliatum (Fortunato et al., 2014).”

We have deleted these lines.

12) Subsection “Hypoxia-sensing transcription factors evolved independently in other multicellular eukaryotes”: It is not very sensible to give a discussion title to a subject that is very loosely related to your study. Removing or merging this section is suggested.

Agreed, we have merged this short discussion with the previous one.